# Maternal Consumption of Dairy Products during Pregnancy Is Associated with Decreased Risk of Emotional Problems in 5-Year-Olds: The Kyushu Okinawa Maternal and Child Health Study

**DOI:** 10.3390/nu14224713

**Published:** 2022-11-08

**Authors:** Mai Quynh Nguyen, Yoshihiro Miyake, Keiko Tanaka, Shizuka Hasuo, Keiji Takahashi, Yoshitaka Nakamura, Hitomi Okubo, Satoshi Sasaki, Masashi Arakawa

**Affiliations:** 1Food Microbiology and Function Research Laboratories, R&D Division, Meiji Co., Ltd., Tokyo 192-0919, Japan; 2Department of Epidemiology and Public Health, Ehime University Graduate School of Medicine, Matsuyama 791-0295, Japan; 3Integrated Medical and Agricultural School of Public Health, Ehime University, Matsuyama 791-0295, Japan; 4Research Promotion Unit, Translational Research Center, Ehime University Hospital, Matsuyama 791-0295, Japan; 5Center for Data Science, Ehime University, Matsuyama 790-8577, Japan; 6Food Science and Technology Research Laboratories, R&D Division, Meiji Co., Ltd., Tokyo 192-0919, Japan; 7Japan Environment and Children’s Study Programme Office, National Institute for Environmental Studies, Osaka 305-8506, Japan; 8Japan Society for the Promotion of Science, Tokyo 102-0083, Japan; 9Department of Social and Preventive Epidemiology, School of Public Health, The University of Tokyo, Tokyo 192-0397, Japan; 10Wellness Research Fields, Faculty of Global and Regional Studies, University of the Ryukyus, Tomigusuku 903-0213, Japan; 11The Department of Cross Cultural Studies, Osaka University of Tourism, Tomigusuku 590-0493, Japan

**Keywords:** behavioral problems, cow’s milk, dairy, Japanese children, maternal intake, pregnancy

## Abstract

Milk is a good source of fats, minerals, and vitamins. The present prebirth cohort study examined the association between maternal dairy product intake during pregnancy and the risk of childhood behavioral problems in 5-year-old Japanese children. Study subjects were 1199 mother–child pairs. Dietary intake was assessed using a diet history questionnaire. Emotional problems, conduct problems, hyperactivity problems, peer problems, and low prosocial behavior were assessed using the parent-reported version of the Strengths and Difficulties Questionnaire. Adjustments were made for *a priori* selected non-dietary confounders and potentially related dietary factors. A significant inverse exposure–response association was observed between maternal total dairy intake during pregnancy and the risk of childhood emotional problems (adjusted odds ratio [OR] between extreme quartiles, 0.62; 95% confidence interval [CI]: 0.36–1.03, *p* for trend, 0.04). The greater maternal consumption of cow’s milk, but not yogurt or cheese, during pregnancy was independently related to a reduced risk of emotional problems in children (adjusted OR between extreme quartiles, 0.41; 95% CI: 0.23–0.70, *p* for trend, 0.003). Higher maternal consumption levels of total dairy products, especially cow’s milk, during pregnancy may be associated with a decreased risk of emotional problems in 5-year-old children.

## 1. Introduction

Maternal nutrition during pregnancy influences fetal development and the health of the child. Mammalian milk is a complex bioactive food and an important vehicle for delivering essential nutrients and endocrine signals to the newborn [1]. On average, cow’s milk is composed of 87% water, 4% to 5% lactose, 3% protein, 3% to 4% fat, 0.8% minerals, and 0.1% vitamins, and the fat fraction is composed of 70% saturated fatty acids and 30% unsaturated fatty acids. Milk is a good source of minerals such as calcium, phosphorus, magnesium, and zinc, and the milk vitamin profile includes liposoluble (A, D, E) and hydrosoluble (B complex and vitamin C) vitamins [2]. In Japan, widely distributed products on the market have a milk fat content of 3.6% or more, and some are fortified with nutrients such as vitamin D and folic acid [3]. Vitamin D deficiency has been observed in Japanese pregnant women [4], and folic acid supplementation during pregnancy is recommended [5]. Some milk products are an excellent source of these nutrients. The National Health and Nutrition Survey in Japan indicated that the average daily per capita intake of dairy products was 117 g among Japanese women aged 20 years or older [6].

Previously, using the parent version of the Strengths and Difficulties Questionnaire (SDQ) for child behavior assessment, we showed that higher maternal intake levels of vitamin B_2_ [7] and calcium [8] were associated with a reduced risk of emotional problems in 5-year-old Japanese children. Maternal intake of monounsaturated fatty acids, α-linolenic acid, and linoleic acid during pregnancy was positively related to childhood emotional problems [9]. Additionally, inverse relationships were observed between maternal vitamin C intake during pregnancy and childhood conduct problems [10], between maternal intake of vitamin B6 [7], vitamin C [10], calcium [8], and magnesium [11] during pregnancy and childhood hyperactivity problems as well as between maternal intake of folate, vitamin B6 [7], and vitamin C [10] during pregnancy and childhood low prosocial behavior. To the best of our knowledge, no epidemiological investigation has examined the association between maternal dairy product intake during pregnancy and the risk of childhood behavioral problems. Research on the possible role of dietary factors in childhood behavioral problems is important because diet is modifiable. In the present prebirth cohort study in Japan, we investigated the association between maternal intake of total dairy products, milk, yogurt, and cheese during pregnancy and the risk of behavioral problems in the children at 5 years of age using data from the Kyushu Okinawa Maternal and Child Health Study (KOMCHS). We hypothesized that higher maternal intake of dairy products during pregnancy would be related to a reduced risk of childhood behavioral problems.

## 2. Materials and Methods

### 2.1. Study Population

The KOMCHS is a prospective prebirth cohort study that was designed to clarify the risk and preventive factors for maternal and child health problems, such as allergy, developmental disorders, and depressive symptoms. The details of the baseline KOMCHS survey were previously described [12]. In the baseline survey, eligible subjects were women in one of the seven prefectures on Kyushu Island in southern Japan, which has a total population of approximately 13.26 million, or in Okinawa Prefecture, which has a population of nearly 1.37 million, who were pregnant between April 2007 and March 2008. We asked 423 obstetric hospitals in the eight prefectures to provide information leaflets explaining the KOMCHS, an application form to participate in the KOMCHS, and a self-addressed stamped return envelope to as many pregnant women as possible. Pregnant women who intended to participate in the KOMCHS mailed the application form to the data management center. Using the contact information in this form, research technicians gave each pregnant woman a detailed explanation of the KOMCHS by telephone and sent a self-administered questionnaire after obtaining verbal consent. Ultimately, 1757 pregnant women between 5 and 39 weeks of gestation provided written informed consent to participate in the KOMCHS and answered a self-administered questionnaire in the baseline survey. Among these 1757 pregnant women, 1590, 1527, 1430, 1362, 1305, 1264, and 1201 mother–child pairs participated in the second (post-delivery), third (approximately 4 months postpartum), fourth (approximately 12 months postpartum), fifth (approximately 24 months postpartum), sixth (approximately 36 months postpartum), seventh (approximately 48 months postpartum), and eighth (approximately 60 months postpartum) surveys, respectively. Two pairs were excluded due to the lack of data on household income. The final analyses comprised 1199 mother–child pairs. The KOMCHS study protocol was approved by the ethics committees at the Faculty of Medicine, Fukuoka University and the Ehime University Graduate School of Medicine.

### 2.2. Measurements

For each survey, study participants completed self-administered questionnaires and mailed them to the data management center. The research technicians then completed missing data or clarified illogical data through telephone interviews. The first part of the questionnaire at baseline elicited information on maternal age, gestation, region of residence, number of children, maternal and paternal educational levels, household income, and depressive symptoms. Depressive symptoms were assessed using a Japanese version of the Center for Epidemiologic Studies Depression Scale (CES-D) [13]. The total CES-D score ranged from 0 to 60, and depressive symptoms were defined as being present when a subject had a CES-D score ≥ 16 [14]. The second part of the questionnaire at baseline was a semi-quantitative, comprehensive diet history questionnaire (DHQ) to assess dietary habits during the preceding month [15,16,17,18,19,20,21]. Estimates of daily food intake (among 150 foods), energy, and selected nutrients were calculated using an ad hoc computer algorithm for the DHQ based on the Standard Tables of Food Composition in Japan [3]. The total intake of dairy products was defined as the sum of cow’s milk, yogurt, and cheese intake; in Japan, yogurt and cheese are produced using mainly cow’s milk. Energy-adjusted intake calculated using the residual method was used for the analyses [22]. The questionnaire in the second survey gathered data regarding the infant’s sex, birth weight, date of birth, and maternal smoking status during pregnancy. Information on household smoking and breastfeeding duration was obtained from the questionnaire until the fourth survey. In the eighth survey, children’s behavioral problems were assessed using a Japanese parent-reported version of the SDQ, which was designed to assess the behavior and emotions of 3- to 16-year-old children [23]. The SDQ consists of five scales: an emotional problems scale, a conduct problems scale, a hyperactivity problems scale, a peer problems scale, and a prosocial scale. These scales were scored using five items each, resulting in 25 items in total. Each item was rated using the following three-point scale: “not true” (0); “somewhat true” (1); and “certainly true” (2). Positively worded items were reverse-scored. The items on each scale were added together to generate a score from 0 to 10. A high score on the prosocial scale reflects strengths, whereas high scores on the other four scales indicate difficulties. These scale scores were then categorized as normal, borderline, or abnormal on the basis of cut-off points that had previously been reported in a sample of Japanese children [24].

### 2.3. Statistical Analysis

The five scale scores were dichotomized, comparing children with borderline and abnormal scores to children with normal scores. We defined emotional problems, conduct problems, hyperactivity problems, peer problems, or low prosocial behavior as present when a child had a borderline or abnormal score in the respective scale (>3, >3, >5, >3, and <6, respectively). Intake of each of the dietary factors under study was categorized at quartile points on the basis of its distribution among the 1199 mothers. A priori, we selected maternal age, gestation at baseline, region of residence at baseline, number of children at baseline, maternal and paternal education, household income, maternal depressive symptoms during pregnancy, maternal alcohol intake during pregnancy, maternal smoking during pregnancy, child’s birth weight, child’s sex, secondhand smoke exposure at home during the first year of life, and breastfeeding duration as potential non-dietary confounding factors. We used maternal age, gestation, and birth weight as continuous variables. The potential dietary confounding factors that were significantly associated with any childhood behavioral problems in the present population in our previous papers were additionally adjusted as follows: citrus fruit was adjusted for in analyses of the associations with emotional problems and conduct problems [10]; vegetables other than green and yellow vegetables, total fruits [10], and total soy products [25] were adjusted for in analyses of the associations with hyperactivity problems; and total vegetables were adjusted for in analyses of the associations with low prosocial behavior [10]. Multiple logistic regression analysis was conducted to estimate the adjusted odds ratios (ORs) and 95% confidence intervals (CIs) for each behavioral problem according to the intake quartiles for total dairy, cow’s milk, yogurt, and cheese, and the lowest quartile was used as the reference. The trend of association was assessed using a logistic regression model assigning consecutive integers (1 to 4) to the intake quartiles for total dairy, cow’s milk, yogurt, and cheese, and two-sided *p* values less than 0.05 were considered statistically significant. All computations were performed using the SAS software package version 9.4 (SAS Institute, Inc., Cary, NC, USA).

## 3. Results

Among the 1199 children who were 59 to 71 months of age, the prevalence values of emotional problems, conduct problems, hyperactivity problems, peer problems, and low prosocial behavior were 12.9%, 19.4%, 13.1%, 8.6%, and 29.2%, respectively. The median maternal age and gestation at baseline were 32.0 (interquartile range [IQR], 29.0–34.0) years and 17.0 (IQR, 14.0–21.0) weeks, respectively (Table 1). Median maternal daily consumption of total energy during pregnancy was 7127 (IQR, 6117–8465) kJ, and median daily energy-adjusted intake levels of total dairy, cow’s milk, yogurt, and cheese during pregnancy were 120.2 g (IQR, 60.6–201.6), 78.3 g (IQR, 31.4–147.7), 22.3 g (IQR, 10.1–52.1), and 3.2 g (IQR, 1.4–6.1), respectively. Maternal cow’s milk consumption during pregnancy was positively associated with gestation at baseline and inversely associated with maternal depressive symptoms during pregnancy and maternal daily total energy consumption.

After adjustment for potential non-dietary and dietary confounding factors, a significant inverse exposure–response association was observed between maternal total dairy intake during pregnancy and the risk of childhood emotional problems; however, the adjusted OR between extreme quartiles fell just short of the significance level (0.62; 95% CI: 0.36–1.03, *p* for trend = 0.04; Table 2). Compared with maternal consumption of cow’s milk during pregnancy in the first quartile, consumption in the fourth quartile was independently related to a reduced risk of emotional problems in children and the inverse linear trend was statistically significant (adjusted OR between extreme quartiles, 0.41; 95% CI: 0.23–0.70; *p* for trend = 0.003). Maternal intake levels of yogurt and cheese during pregnancy were not related to the risk of childhood emotional problems. There were no measurable associations between maternal consumption of total dairy, cow’s milk, yogurt, or cheese during pregnancy and childhood conduct, hyperactivity, or peer problems or low prosocial behavior; however, a significant positive exposure–response relationship was found between maternal yogurt consumption during pregnancy and the risk of low prosocial behavior (*p* for trend = 0.04).

Maternal consumption of cow’s milk during pregnancy was significantly correlated with maternal intake of calcium and vitamin B_2_ during pregnancy (Pearson’s correlation coefficient: 0.76, *p* < 0.0001 and 0.47, *p* < 0.0001, respectively). After additional adjustment for maternal intake of calcium or vitamin B_2_ during pregnancy as continuous variables, an inverse association between maternal cow’s milk intake during pregnancy and childhood emotional problems remained significant (adjusted OR, 0.40 [95% CI: 0.20–0.81; *p* for trend = 0.04] and 0.44 [95% CI: 0.24–0.79; *p* for trend = 0.02], respectively).

## 4. Discussion

To the best of our knowledge, the present pre-birth cohort study was the first to demonstrate the independent inverse association of higher maternal total dairy, especially cow’s milk, intake during pregnancy with the risk of emotional problems in children. An Australian longitudinal study of 1554 mother–child pairs showed a significant inverse association between maternal pre-pregnancy dairy consumption and total behavioral difficulties, which included the following four scales that were based on the SDQ in children aged 5–12 years: hyperactivity, emotional, conduct, and peer problems. However, there was no information on the relationship between maternal pre-pregnancy dairy consumption and childhood emotional problems [27]. In a cross-sectional study of 986 Korean children aged 8–11 years, intake of dairy products was significantly inversely associated with delinquent behavior, aggressive behavior, and externalizing problems based on the Child Behavioral Checklist, although the association between dairy product intake and emotional problems was not examined [28]. With regard to the beneficial association between dairy intake and childhood behavioral problems, these findings are in partial agreement with our results.

Because an inverse association between maternal cow’s milk intake during pregnancy and childhood emotional problems persisted after further adjustment for maternal calcium or vitamin B_2_ intake during pregnancy, the inverse association is unlikely to be ascribed to the calcium or vitamin B_2_ in cow’s milk. Milk fat globule membrane (MFGM) and milk proteins might be particularly important. Postnatal supplementation of ganglioside- and phospholipid-enriched complex-milk-lipids beta serum concentrate improved learning and memory in rats [29]. Formula with bovine MFGM promoted reflex development and changed brain phospholipid and metabolite composition in rats [30]. A randomized controlled trial demonstrated that infants fed an MFGM-supplemented experimental formula between 2 and 6 months of age performed better on cognitive testing at 12 months compared with infants fed standard formula and at a level not significantly different from the breastfed reference group [31]. When the diet of postnatal piglets was supplemented with lactoferrin, an iron-binding milk glycoprotein, from days 3 to 38, lactoferrin supplementation promoted early neurodevelopment and cognition by upregulating the brain-derived neurotrophic factor signaling pathway and polysialylation [32]. A recent study in chronic stress model mice has revealed the potential of milk casein to prevent stress-induced brain dysfunction and anxiety-like behavior [33]. Given the protective effects of MFGM or any milk proteins on neurodevelopment in early childhood, higher maternal intake of cow’s milk during pregnancy might provide persistent beneficial effects against the development of emotional problems in children. The observed positive exposure–response relationship between maternal yogurt intake during pregnancy and childhood low prosocial behavior might have been observed by chance. On the other hand, regarding the observed inverse association between maternal cow’s milk intake during pregnancy and childhood emotional problems, the *p* value for adjusted odds ratio between extreme quartiles was 0.0015 and the *p* for the trend was 0.003; the α error is very small, and thus, this finding is not likely to be a chance phenomenon.

Methodological strengths of the present study included its prebirth prospective cohort design with a relatively large sample size, its relatively long duration of follow-up that included a period from prenatal life to 5 years of age, and its extensive information on potential confounding factors.

There are several limitations of the current study. Although adjustment was attempted for a variety of potential confounding factors, the possibility of residual confounding cannot be ignored. The DHQ could only give approximate consumption and was designed to assess dietary intake for 1 month before the questionnaire was completed. Additionally, the DHQ was completed anywhere between week 5 and 39 of pregnancy. Thus, any non-differential exposure misclassification would have biased the magnitude of the observed associations, making our estimate more conservative than the true effect size.

The SDQ data were reported by parents, which may have introduced bias, and it is uncertain whether the dichotomization cut-off points for the five outcomes under study were reasonable, although they were chosen based on a previous study conducted in Japan [24]. Non-differential outcome misclassification might have altered the observed association toward the null.

Among the 1757 pregnant women who completed the baseline survey between April 2007 and March 2008, 556 mother–child pairs did not participate in the eighth survey. No evident differences were observed between the 556 non-participants and the 1201 participants in the eighth survey regarding the distribution for the number of children, depressive symptoms during pregnancy, or alcohol intake during pregnancy. Participants in the eighth survey were more likely to be older, have participated in the baseline survey earlier in gestation, live in Fukuoka Prefecture, report high maternal and paternal educational levels, and have a high household income compared with those of the non-participants. We were not able to calculate the participation rate at baseline because the number of pregnant women who received the leaflet explaining the KOMCHS, an application form, and a set of self-addressed and stamped return envelope at the 423 collaborating obstetric hospitals was unknown. Among the 1757 participating mothers at baseline, 978 lived in Fukuoka Prefecture. Data collected by the government of Fukuoka Prefecture showed that there were 46,393 children born in 2007 and 46,695 children born in 2008, which suggests that the participation rate in the baseline survey of the KOMCHS must have been low. Thus, our participants were probably not representative of Japanese women in the general population. For example, a population census conducted in 2000 in Fukuoka Prefecture showed that the proportions of women aged 30–34 years with< 13 years, 13–14 years, ≥15 years, and an unknown number of years of education were 52.0%, 31.5%, 11.8%, and 4.8%, respectively [34]. The corresponding figures for our study were 20.9%, 33.3%, 45.9%, and 0.0%, respectively.

## 5. Conclusions

The present prebirth cohort study suggested that a higher maternal consumption level of total dairy products, especially cow’s milk, during pregnancy was independently associated with a decreased risk of emotional problems in 5-year-old children. We acknowledge the necessity of additional epidemiological studies and studies investigating the mechanisms underlying the observed preventive association. However, dietary modification to increase maternal intake of total dairy products and cow’s milk during pregnancy may be an important strategy to prevent childhood emotional problems.

## Figures and Tables

**Table 1 nutrients-14-04713-t001:** Characteristics of 1199 mother–child pairs, according to quartile (Q) of maternal cow’s milk intake during pregnancy ^a^.

	Total (*n* = 1199) ^b^	Cow’s Milk ^c^				
		Q1	Q2	Q3	Q4	*p* for Trend ^d^
Baseline characteristics						
Maternal age, years	32.0 (29.0–34.0)	32.0	31.0	32.0	32.0	0.43
Gestation, weeks	17.0 (14.0–21.0)	17.0	17.0	18.0	18.0	0.0001
Region of residence, %						0.55
Fukuoka Prefecture	57.8	58.5	58.7	57.0	57.0	
Other than Fukuoka Prefecture in Kyushu	32.8	30.8	33.3	33.7	33.3	
Okinawa Prefecture	9.4	10.7	8.0	9.3	9.7	
Number of living children born to the same mother, %						0.06
0	40.4	34.5	40.3	43.3	43.3	
1	40.0	44.5	39.7	38.7	37.3	
≥2	19.6	21.1	20.0	18.0	19.3	
Maternal education, years, %						0.54
13	20.9	24.4	18.0	22.0	19.0	
13–14	33.3	32.8	30.7	34.0	35.7	
≥15	45.9	42.8	51.3	44.0	45.3	
Paternal education, years, %						0.76
<13	28.6	29.1	29.7	29.3	26.3	
13–14	14.4	12.7	14.3	14.7	16.0	
≥15	57.0	58.2	56.0	56.0	57.7	
Household income, yen/year, % ^e^						0.45
<4,000,000	32.2	33.8	30.3	32.7	32.0	
4,000,000–5,999,999	37.5	38.5	39.0	36.0	36.3	
≥6,000,000	30.4	27.8	30.7	31.3	31.7	
Maternal depressive symptoms during pregnancy, %	18.2	22.4	18.0	16.7	15.7	0.03
Maternal alcohol intake during pregnancy, %	13.2	13.4	11.0	15.3	13.0	0.71
Maternal daily intake						
Total energy, kJ	7127 (6117–8465)	7574	6466	7476	7155	0.04
Total dairy, g ^f^	120.2 (60.6–201.6)					
Cow’s milk, g ^f^	78.3 (31.4–147.7)					
Yogurt, g ^f^	22.3 (10.1–52.1)					
Cheese, g ^f^	3.2 (1.4–6.1)					
Total vegetables, g ^f^	187.2 (139.0–264.4)					
Vegetables other than green and yellow vegetables, g ^f^	111.2 (79.3–158.9)					
Total fruits, g ^f^	125.3 (80.8–185.8)					
Citrus fruit, g ^f^	19.7 (7.0–46.2)					
Total soy products, g ^f^	46.9 (30.0–70.2)					
Calcium, mg ^f^	481.6 (398.2–587.0)					
Vitamin B2, mg ^f^	1.3 (1.1–1.5)					
Characteristics at the postnatal assessment						
Maternal smoking during pregnancy, %	7.3	10.0	7.3	4.7	7.0	0.08
Birth weight, g	3012 (2772–3246)	3034	3026	3020	2960	0.11
Male sex, %	47.4	47.2	47.7	45.0	49.7	0.71
Breastfeeding duration, months, %						0.50
<6	10.8	11.7	11.3	10.0	10.3	
≥6	89.2	88.3	88.7	90.0	89.7	
Smoking in the household during the first year of life, %	27.4	30.8	25.7	29.7	23.7	0.13

^a^ Values are medians for continuous variables and percentages of subjects for categorical variables. ^b^ Values are medians (interquartile ranges) for continuous variables and percentages of subjects for categorical variables. ^c^ The intake range of each quartile was as follows: Q1 (<31.4 g), Q2 (31.4 to <78.3 g), Q3 (78.3 to <147.7 g), Q4 (≥147.7 g). ^d^ For continuous variables, a linear trend test was used; for categorical variables, a Mantel–Haenszel χ^2^-test was used. ^e^ Average household income was 5,562,000 yen in 2007 [26]. ^f^ Food intake level was adjusted for total energy intake using the residual method.

**Table 2 nutrients-14-04713-t002:** Odds ratios (ORs) and 95% confidence intervals (CIs) for behavioral problems assessed by the Strength and Difficulties Questionnaire in 1199 5-year-old children by quartiles of maternal dairy product intake during pregnancy.

	Emotional Problems		Conduct Problems		Hyperactivity Problems		Peer Problems		Low Prosocial Behavior	
Variables ^a^	Risk (%)	Adjusted OR (95% CI) ^b^	Risk (%)	Adjusted OR (95% CI) ^b^	Risk (%)	Adjusted OR (95% CI) ^c^	Risk (%)	Adjusted OR (95% CI) ^d^	Risk (%)	Adjusted OR (95% CI) ^e^
Total dairy										
Q1 (32.7)	14.4	1.00	18.7	1.00	13.4	1.00	8.4	1.00	26.1	1.00
Q2 (87.8)	15.3	1.03 (0.64–1.66)	19.7	1.09 (0.71–1.66)	13.0	0.89 (0.54–1.47)	7.7	0.95 (0.51–1.74)	33.0	1.37 (0.95–1.98)
Q3 (156.7)	11.7	0.78 (0.47–1.28)	19.3	1.19 (0.78–1.83)	13.0	0.98 (0.59–1.61)	8.3	1.11 (0.61–2.02)	25.7	1.00 (0.68–1.46)
Q4 (250.6)	10.3	0.62 (0.36–1.03)	20.0	1.24 (0.81–1.91)	13.0	0.92 (0.56–1.51)	10.0	1.29 (0.73–2.33)	32.0	1.34 (0.92–1.94)
*p* for trend		0.04		0.28		0.85		0.32		0.36
Cow’s milk										
Q1 (9.3)	15.7	1.00	20.1	1.00	14.7	1.00	8.0	1.00	30.4	1.00
Q2 (49.4)	14.3	0.92 (0.57–1.47)	18.0	0.96 (0.63–1.46)	11.0	0.68 (0.41–1.11)	8.0	1.06 (0.58–1.94)	29.0	0.93(0.64–1.33)
Q3 (112.9)	14.3	0.88 (0.55–1.40)	20.3	1.14 (0.75–1.73)	14.7	1.02 (0.64–1.64)	8.7	1.15 (0.63–2.10)	28.0	0.90 (0.62–1.29)
Q4 (190.5)	7.3	0.41 (0.23–0.70)	19.3	1.10 (0.72–1.68)	12.0	0.77 (0.47–1.26)	9.7	1.33 (0.74–2.41)	29.3	0.93 (0.65–1.35)
*p* for trend		0.003		0.49		0.63		0.32		0.69
Yogurt										
Q1 (4.0)	11.4	1.00	18.7	1.00	12.7	1.00	8.4	1.00	27.4	1.00
Q2 (15.2)	12.7	1.13 (0.67–1.90)	19.3	1.09 (0.71–1.67)	12.3	0.97 (0.58–1.63)	8.3	1.02 (0.56–1.86)	25.7	0.92 (0.63–1.35)
Q3 (34.6)	13.0	1.20 (0.71–2.01)	19.3	1.17 (0.76–1.80)	13.0	1.07 (0.65–1.78)	8.3	1.05 (0.58–1.91)	32.7	1.40 (0.97–2.02)
Q4 (81.5)	14.7	1.38 (0.83–2.33)	20.3	1.40 (0.91–2.16)	14.3	1.28 (0.77–2.13)	9.3	1.19 (0.66–2.17)	31.0	1.31 (0.90–1.91)
*p* for trend		0.21		0.12		0.31		0.56		0.04
Cheese										
Q1 (0.1)	16.4	1.00	21.4	1.00	11.7	1.00	9.0	1.00	28.8	1.00
Q2 (2.3)	10.3	0.59 (0.35–0.97)	19.3	0.92 (0.61–1.40)	14.7	1.37 (0.83–2.26)	8.3	0.91 (0.51–1.64)	27.0	0.95 (0.65–1.37)
Q3 (4.1)	14.3	0.88 (0.55–1.41)	21.7	1.06 (0.70–1.60)	15.3	1.37 (0.84–2.27)	7.3	0.82 (0.44–1.49)	29.0	0.95 (0.65–1.37)
Q4 (11.7)	10.7	0.65 (0.39–1.06)	15.3	0.71 (0.45–1.09)	10.7	0.92 (0.54–1.56)	9.7	1.14 (0.64–2.03)	32.0	1.16 (0.80–1.67)
*p* for trend		0.24		0.21		0.78		0.74		0.45

CI, confidence interval; OR, odds ratio; Q, quartile; ^a^ Quartile medians in g/day adjusted for energy intake using the residual method are given in parentheses. ^b^ Adjustment for maternal age, gestation at baseline, region of residence at baseline, number of children at baseline, maternal and paternal education, household income, maternal depressive symptoms during pregnancy, maternal alcohol intake during pregnancy, maternal smoking during pregnancy, child’s birth weight, child’s sex, postnatal secondhand smoke exposure at home during the first year of life, breastfeeding duration, and citrus fruit consumption during pregnancy. ^c^ Adjustment for maternal age, gestation at baseline, region of residence at baseline, number of children at baseline, maternal and paternal education, household income, maternal depressive symptoms during pregnancy, maternal alcohol intake during pregnancy, maternal smoking during pregnancy, child’s birth weight, child’s sex, postnatal secondhand smoke exposure at home during the first year of life, breastfeeding duration, and consumption of vegetables other than green and yellow vegetables, total fruits, and total soy products during pregnancy. ^d^ Adjustment for maternal age, gestation at baseline, region of residence at baseline, number of children at baseline, maternal and paternal education, household income, maternal depressive symptoms during pregnancy, maternal alcohol intake during pregnancy, maternal smoking during pregnancy, child’s birth weight, child’s sex, postnatal secondhand smoke exposure at home during the first year of life, and breastfeeding duration. ^e^ Adjustment for maternal age, gestation at baseline, region of residence at baseline, number of children at baseline, maternal and paternal education, household income, maternal depressive symptoms during pregnancy, maternal alcohol intake during pregnancy, maternal smoking during pregnancy, child’s birth weight, child’s sex, postnatal secondhand smoke exposure at home during the first year of life, breastfeeding duration, and total vegetable consumption during pregnancy.

## Data Availability

The datasets analyzed during the current study are not publicly available because the steering committee and the participants did not approve unrestricted data sharing.

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
