# Peer review of "Maternal Consumption of Dairy Products during Pregnancy Is Associated with Decreased Risk of Emotional Problems in 5-Year-Olds: The Kyushu Okinawa Maternal and Child Health Study"

_nutrients, 2022, doi:10.3390/nu14224713_

Round 1

Reviewer 1 Report

Dear authors, this is a very interesting study on how the mother's diet, especially dairy products, can influence the subsequent development of the child.

I have some questions and suggestions.

- Dairy products can be different in composition and properties depending on the animal origin (cow, goat, buffalo, sheep); did you somehow make a differentiation of this kind? Or did all mothers consume only cow's milk?

- What do you mean by "maternal depressive symptoms during pregnancy" and how they were quantified?

-  The feeding the child in the first years of life can also influence the social and emotional behavior of the child; in this sense, maybe the study should be continued and see if there are others confounding factors to be taken into account. 

- The same as above is also valid for prematurity or associated congenital anomalies; were these factors analyzed?

- The Discussion chapter should be more extensive, the authors should discuss more about the results obtained in comparison with other similar studies. They should also discuss the physio-pathological mechanism involved in the production of such effects discovered by their study.

Author Response

Reviewer 1 Comments for the Author...

Dear authors, this is a very interesting study on how the mother's diet, especially dairy products, can influence the subsequent development of the child.

I have some questions and suggestions.

Response:

Thank you for your careful review and insightful comments. We carefully revised the manuscript in accordance with your suggestions.

- Dairy products can be different in composition and properties depending on the animal origin (cow, goat, buffalo, sheep); did you somehow make a differentiation of this kind? Or did all mothers consume only cow's milk?

Response:

For yogurt and cheese, the differentiation related to the animal of origin was not addressed in the Diet History Questionnaire. However, Japanese dairy products (e.g., yogurt and cheese) are produced using mainly cow’s milk. The sentence “The total intake of dairy products was defined as the sum of cow’s milk, yogurt, and cheese intake; in Japan, yogurt and cheese are produced using mainly cow’s milk.” was added to the Materials and Methods section (Page 3 Lines 113–115 in the revised manuscript).

- What do you mean by "maternal depressive symptoms during pregnancy" and how they were quantified?

Response:

Maternal depressive symptoms during pregnancy were defined using information from the Center for Epidemiologic Studies Depression Scale (CES-D) that was obtained in the baseline survey.

The words “maternal depressive symptoms during pregnancy” in the Materials and Methods section were changed to “depressive symptoms” (Page 3 Lines 105–106 in the revised manuscript).

The sentence “Depressive symptoms were assessed using a Japanese version of the Center for Epidemiologic Studies Depression Scale (CES-D) [13]. The total CES-D score ranged from 0 to 60, and depressive symptoms were defined as being present when a subject had a CES-D score ≥16 [14]” was added to the Materials and Methods section (Page 3 Lines 106–109 in the revised manuscript).

The following references, which are mentioned in the above sentences, were added:

  1. Shima, S.; Shikano, T.; Kitamura, T.; Asai, M. New self-rated scale for depression (in Japanese). Jpn J Clin Psychiatry. 1985, 27, 717-723.
  2. Radloff, LS. The CES-D scale: a self-report depression scale for research in the general population. Appl Psychol Meas. 1977, 1, 385-401.

-  The feeding the child in the first years of life can also influence the social and emotional behavior of the child; in this sense, maybe the study should be continued and see if there are others confounding factors to be taken into account.

Response:

We plan to examine the association of postnatal feeding with childhood behavioral problems in a subsequent study. We did not add any text concerning this issue.

- The same as above is also valid for prematurity or associated congenital anomalies; were these factors analyzed?

Response:

For prematurity, our analysis adjusted for the child’s birth weight. There were few children with congenital anomalies in our population, and thus, we did not adjust for congenital anomalies. We did not add any text concerning this issue.

- The Discussion chapter should be more extensive, the authors should discuss more about the results obtained in comparison with other similar studies. They should also discuss the physio-pathological mechanism involved in the production of such effects discovered by their study.

Response:

The phrases “which included the following four scales that were based on the SDQ in children aged 5–12 years: hyperactivity, emotional, conduct, and peer problems.”, “However, there was no information on the relationship between maternal pre-pregnancy dairy consumption and childhood emotional problems”, “delinquent behavior, aggressive behavior, and”, “although the association between dairy product intake and emotional problems was not examined”, and “With regard to the beneficial association between dairy intake and childhood behavioral problems” were added to the Discussion section (Page 7 Lines 226–234 in the revised manuscript).

The sentence “Given the protective effects of MFGM or any milk proteins on neurodevelopment in early childhood, higher maternal intake of cow’s milk during pregnancy might provide persistent beneficial effects against the development of emotional problems in children.” was added to the Discussion section (Page 7 Lines 251–254 in the revised manuscript)

Reviewer 2 Report

Thank you that you have given me opportunity to review this manuscript Maternal consumption of dairy products during pregnancy is associated with decreased risk of emotional problems in 5-year-olds: the Kyushu Okinawa Maternal and Child Health Study

The topic of the manuscript is very interesting. The Authors have done good work and presented clear sufficient data.  The methods are adequately described. The results have clearly presented. The authors used appropriate statistical methods. The conclusions are consistent with the presented evidence and arguments.

Only some remarks:

Introduction:

The introduction does not describe what milk is available in Japan, what is the composition of milk that pregnant women could drink. Is it fat milk or not?

Are Japanese pregnant women not tested for vitamin D3 levels and folic acid supplementation?

Please add 1-2 sentences.

Results:

In the results, the authors present, for example, household income, yen / year,% <4,000,000. For a person from outside Japan, it is difficult to judge whether this is enough money for a comfortable life, or whether she is a very poor person. It is worth referring to the national average earnings, and this sum of money for what is enough.

 Please explain it.

Author Response

Reviewer 2 Comments for the Author...

Thank you that you have given me opportunity to review this manuscript Maternal consumption of dairy products during pregnancy is associated with decreased risk of emotional problems in 5-year-olds: the Kyushu Okinawa Maternal and Child Health Study

The topic of the manuscript is very interesting. The Authors have done good work and presented clear sufficient data. The methods are adequately described. The results have clearly presented. The authors used appropriate statistical methods. The conclusions are consistent with the presented evidence and arguments.

Only some remarks:

Response:

Thank you for your careful review and insightful comments. We have carefully revised the manuscript in accordance with your remarks.

Introduction:

The introduction does not describe what milk is available in Japan, what is the composition of milk that pregnant women could drink. Is it fat milk or not?

Are Japanese pregnant women not tested for vitamin D3 levels and folic acid supplementation?

Please add 1-2 sentences.

Response:

The sentences “In Japan, widely distributed products on the market have a milk fat content of 3.6% or more, and some are fortified with nutrients such as vitamin D and folic acid [3]. Vitamin D deficiency has been observed in Japanese pregnant women [4], and folic acid supplementation during pregnancy is recommended [5]. Some milk products are an excellent source of these nutrients.” were added to the Introduction (Page 2 Lines 49–53 in the revised manuscript).

The following references, mentioned in the above sentences, were added:

  1. Science and Technology Agency. Standard tables of food composition in Japan, Fifth revised and enlarged edition (in Japanese). Printing Bureau of the Ministry of Finance: Tokyo, Japan, 2005.
  2. Kanatani, K.T.; Nakayama, T.; Adachi, Y.; Hamazaki, K.; Onishi, K.; Konishi, Y.; Kawanishi, Y.; Go, T.; Sato, K.; Kurozawa, Y.; Inadera, H.; Konishi, I.; Sasaki, S.; Oyama, H.; Japan Environment and Children's Study Group. High frequency of vitamin D deficiency in current pregnant Japanese women associated with UV avoidance and hypo-vitamin D diet. PLoS One. 2019, 4, 14(3), e0213264. https://doi.org/10.1371/journal.pone.0213264.
  3. Ministry of Health, Labour and Welfare of Japan. Dietary reference intakes for Japanese, 2020 Edition, https://www.mhlw.go.jp/stf/seisakunitsuite/bunya/kenkou_iryou/kenkou/eiyou/syokuji_kijyun.html (in Japanese)

Results:

In the results, the authors present, for example, household income, yen / year,% <4,000,000. For a person from outside Japan, it is difficult to judge whether this is enough money for a comfortable life, or whether she is a very poor person. It is worth referring to the national average earnings, and this sum of money for what is enough.

 Please explain it.

Response:

The sentence “Average household income was 5,562,000 yen in 2007 [26].” was added to the footnote of Table 1.

The following reference, mentioned in the above sentence, was added:

  1. Ministry of Health, Labour and Welfare of Japan. Overview of the 2008 Comprehensive Survey of Living Conditions https://www.mhlw.go.jp/toukei/saikin/hw/k-tyosa/k-tyosa08/2-1.html (in Japanese)

Reviewer 3 Report

 No comments 

Author Response

Reviewer 3 Comments for the Author...

No comments

Response:

Thank you for reviewing our paper.

Reviewer 4 Report

In this paper the authors want to sent the message that the consumption of cow's milk by the mother during pregnancy has a positive effect on psychological items of the ofspring at 6 years of age.

I do have some questions and remarks.

1. Was the question if cow's milk consumed by the mother during pregnancy has an effect on psychological outcome at 6 years an a priori question raised by the onset of the study or is this a post-hoc analysis? Did you collect cord blood to study differences in protein or fat profile between infants?

2. the only significant finding is a correlation between cow's milk and emotional problems in children of mothers with the highest milk intake, Q4. The values of Q1,2 and 3 were almost the same. I sthis not a chance finding?

3. The hypothesis that cow's milk during pregnancy has an effect on the ofspring is not really supported by other data. Only one study in animals might give an indication, but this was a postnatal supplementation. The few other studies also involve postnatal nutrition.

4. What were the scores of the different psychological scores in the normal population?

5. 13 of the first 14 references are by the group od authors of this paper. They found effects of many food ingredients on the psychological development. Are these results confirmed by other studies?

Author Response

Reviewer 4 Comments for the Author...

In this paper the authors want to sent the message that the consumption of cow's milk by the mother during pregnancy has a positive effect on psychological items of the ofspring at 6 years of age.

I do have some questions and remarks.

Response:

Thank you for your insightful editorial comments. We have carefully revised the manuscript in accordance with your remarks.

  1. Was the question if cow's milk consumed by the mother during pregnancy has an effect on psychological outcome at 6 years an a priori question raised by the onset of the study or is this a post-hoc analysis? Did you collect cord blood to study differences in protein or fat profile between infants?

Response:

The question whether maternal dietary consumption, including dairy products, during pregnancy has an effect on psychological outcome at 5 years was an a priori question in the study. In our study, we did not collect cord blood for analysis. We have not made any revisions in response to this comment.

  1. the only significant finding is a correlation between cow’s milk and emotional problems in children of mothers with the highest milk intake, Q4. The values of Q1,2 and 3 were almost the same. Is this not a chance finding?

Response:

The p value for the adjusted odds ratio between extreme quartiles was 0.0015 and the p for trend was 0.003. The α error is very small, and thus, this finding is not likely to be a chance phenomenon. We did not add any text concerning this issue.

  1. The hypothesis that cow's milk during pregnancy has an effect on the ofspring is not really supported by other data. Only one study in animals might give an indication, but this was a postnatal supplementation. The few other studies also involve postnatal nutrition.

Response:

As mentioned in the text, an Australian longitudinal study of 1554 mother–child pairs showed a significant inverse association between maternal pre-pregnancy dairy consumption and total behavioral difficulties, which consisted of the following four scales based on the SDQ in children aged 5–12 years: hyperactivity, emotional, conduct, and peer problems. We have not made any revisions in response to this comment.

  1. What were the scores of the different psychological scores in the normal population?

Response:

The passage “(>3, >3, >5, >3, and <6, respectively)” was added to the Materials and Methods section (Page 3 Lines 135–136 in the revised manuscript)

  1. 13 of the first 14 references are by the group od authors of this paper. They found effects of many food ingredients on the psychological development. Are these results confirmed by other studies?

Response:

To the best of our knowledge, our previous results have not been confirmed by other studies. We have not made any revisions in response to this comment.

Reviewer 5 Report

Welldone on your study. Please see below my comment on your manuscript.

Introduction

1) There is a difference between writing an abstract and writing a study introduction. Your introduction reads as though it was an abstract. Please rewrite the entire introduction section and while doing so, please include:

a) a overview of the topic e.g. the importance of ensuring appropriate maternal nutrition during pregnancy.

b) some statistics to inform why you decided to conduct the study. This might be local, national or global statistics.

c) a concise literature of available evidence on the topic. What has been reported on this topic from previous studies?

d) from the literature review, the identified gap in evidence.

e) a build-up to the study rationale.

f) the aim of the study.

g) a brief statement on the policy implication of the study.  

Methods

2) In the methods section, you wrote "We asked 423 obstetric hospitals in the eight prefectures to provide information leaflets ex-67 plaining the KOMCHS, an application form to participate in the KOMCHS.....", who composed the information leaflets and application form used for recruitment of participants?

3) Please provide more information about the Kyushu Okinawa Maternal and Child Health Study (KOMCHS) to give some context around the study e.g. study methodology including sampling technique, etc. Simply writing "The details of the baseline 62 KOMCHS survey were previously described [7]" is not enough.

4) Please write out the full meaning of SDQ at first mention in the manuscript.

Discussion

5) Please rewrite the discussion section and use results from other HUMAN studies to support or negate findings from your study. Results from animal studies should only be stated to further substantiate your result but not as sole evidence.

6) What are the policy implications of the study?

Author Response

Reviewer 5 Comments for the Author...

Well done on your study. Please see below my comment on your manuscript.

Response:

Thank you for your insightful editorial comments. We have carefully revised the manuscript in accordance with your remarks.

Introduction

1) There is a difference between writing an abstract and writing a study introduction. Your introduction reads as though it was an abstract. Please rewrite the entire introduction section and while doing so, please include:

a) a overview of the topic e.g. the importance of ensuring appropriate maternal nutrition during pregnancy.

b) some statistics to inform why you decided to conduct the study. This might be local, national or global statistics.

c) a concise literature of available evidence on the topic. What has been reported on this topic from previous studies?

d) from the literature review, the identified gap in evidence.

e) a build-up to the study rationale.

f) the aim of the study.

g) a brief statement on the policy implication of the study.

Response:

The sentence “Maternal nutrition during pregnancy influences fetal development and the health of the child. Mammalian milk is a complex bioactive food and an important vehicle for delivering essential nutrients and endocrine signals to the newborn [1].” was added to the Introduction (Page 1 Lines 42–44 in the revised manuscript).

The sentences “In Japan, widely distributed products on the market have a milk fat content of 3.6% or more, and some are fortified with nutrients such as vitamin D and folic acid [3]. Vitamin D deficiency has been observed in Japanese pregnant women [4], and folic acid supplementation during pregnancy is recommended [5]. Some milk products are an excellent source of these nutrients. The National Health and Nutrition Survey in Japan indicated that the average daily per capita intake of dairy products was 117 g among Japanese women aged 20 years or older [6].” were added to the Introduction (Page 2 Lines 49–55 in the revised manuscript).

The sentence “Research on the possible role of dietary factors in childhood behavioral problems is important because diet is modifiable.” was added to the Introduction (Page 2 Lines 68–69 in the revised manuscript).

The following references, mentioned in the above sentences, were added:

1. Brantsæter, A.L.; Olafsdottir, A.S.; Forsum, E.; Olsen, S.F.; Thorsdottir, I. Does milk and dairy consumption during pregnancy influence fetal growth and infant birthweight? A systematic literature review. Food Nutr Res. 2012, 56. https://doi: 10.3402/fnr.v56i0.20050.

3. Science and Technology Agency. Standard tables of food composition in Japan, Fifth revised and enlarged edition (in Japanese). Printing Bureau of the Ministry of Finance: Tokyo, Japan, 2005.

4. Kanatani, K.T.; Nakayama, T.; Adachi, Y.; Hamazaki, K.; Onishi, K.; Konishi, Y.; Kawanishi, Y.; Go, T.; Sato, K.; Kurozawa, Y.; Inadera, H.; Konishi, I.; Sasaki, S.; Oyama, H.; Japan Environment and Children's Study Group. High frequency of vitamin D deficiency in current pregnant Japanese women associated with UV avoidance and hypo-vitamin D diet. PLoS One. 2019, 4, 14(3), e0213264. https://doi.org/10.1371/journal.pone.0213264.

5. Ministry of Health, Labour and Welfare of Japan. Dietary reference intakes for Japanese, 2020 Edition, https://www.mhlw.go.jp/stf/seisakunitsuite/bunya/kenkou_iryou/kenkou/eiyou/syokuji_kijyun.html (in Japanese)

6. National Institute of Health and Nutrition. The National Health and Nutrition Survey Japan, 2019. (in Japanese) Tokyo: National Institute of Health and Nutrition, 2019.

Methods

2) In the methods section, you wrote "We asked 423 obstetric hospitals in the eight prefectures to provide information leaflets ex-67 plaining the KOMCHS, an application form to participate in the KOMCHS.....", who composed the information leaflets and application form used for recruitment of participants?

Response:

The information leaflets and application form used to recruit participants were prepared by the authors Yoshihiro Miyake and Keiko Tanaka, and they were then distributed to the obstetrics hospitals. We have not made any revisions in response to this comment.

3) Please provide more information about the Kyushu Okinawa Maternal and Child Health Study (KOMCHS) to give some context around the study e.g. study methodology including sampling technique, etc. Simply writing "The details of the baseline 62 KOMCHS survey were previously described [7]" is not enough.

Response:

The first sentence of the Materials and Methods section was changed to “The KOMCHS is a prospective prebirth cohort study that was designed to clarify the risk and preventive factors for maternal and child health problems such as allergy, developmental disorders, and depressive symptoms.” (Page 2 Lines 75–77 in the revised manuscript).

The sentence “Using the contact information in this form, research technicians gave each pregnant woman a detailed explanation of the KOMCHS by telephone and sent her a self-administered questionnaire after obtaining her agreement.” was added to the Materials and Methods section (Page 2 Lines 86–88 in the revised manuscript).

4) Please write out the full meaning of SDQ at first mention in the manuscript.

Response:

SDQ was defined as follows: “Strengths and Difficulties Questionnaire (SDQ)” in the Introduction section (Page 2 Lines 56–57 in the revised manuscript)

Discussion

5) Please rewrite the discussion section and use results from other HUMAN studies to support or negate findings from your study. Results from animal studies should only be stated to further substantiate your result but not as sole evidence.

Response:

In the first paragraph of the Discussion section, results from other studies in humans were described. The following statements including these details were added to the Discussion section: “which included the following four scales that were based on the SDQ in children aged 5–12 years: hyperactivity, emotional, conduct, and peer problems.”, “However, there was no information on the relationship between maternal pre-pregnancy dairy consumption and childhood emotional problems”, “delinquent behavior, aggressive behavior, and”, “although the association between dairy product intake and emotional problems was not examined”, and “With regard to the beneficial association between dairy intake and childhood behavioral problems” (Page 7 Lines 226–234 in the revised manuscript).

In the second paragraph of the Discussion section, possible underlying mechanisms were described using the results from animal studies.

6) What are the policy implications of the study?

Response:

The words “However, dietary modification to increase maternal intake of total dairy products and cow’s milk during pregnancy may be an important strategy to prevent childhood emotional problems.” were added to the Conclusions section (Page 8 Lines 299–301 in the revised manuscript).

Round 2

Reviewer 1 Report

Dear Authors,

Thank you for the answers to my suggestions. I believe that the manuscript has been improved and deserves to be published.

Best wishes!

Author Response

Reviewer 1 Comments for the Author...

Dear Authors,

Thank you for the answers to my suggestions. I believe that the manuscript has been improved and deserves to be published.

Best wishes!

Response:

Thank you for your careful review of our revised manuscript.

Reviewer 4 Report

I thank the authors for their response. The answers however are not answering my concerns. Also, no changes were made in the paper. The conclusions of the article are essentially based on one number. With so many correlations one might expect at least one positive correlation. Secondly, there is very limited physiological support for the finding. Therefore I can not recommend acceptance of the paper.

Author Response

Reviewer 4 Comments for the Author...

I thank the authors for their response. The answers however are not answering my concerns. Also, no changes were made in the paper. The conclusions of the article are essentially based on one number. With so many correlations one might expect at least one positive correlation. Secondly, there is very limited physiological support for the finding. Therefore I can not recommend acceptance of the paper.

Response:

Thank you very much for your insightful editorial comments. We have carefully revised the manuscript according to your remarks.

      According to your comment “Also, no changes were made in the paper.” and your previous #1 comment, the sentence “We hypothesized that higher maternal intake of dairy products during pregnancy would be related to a reduced risk of childhood behavioral problems.” was added to the Introduction (Page 2 Lines 73-74 in the revised manuscript).

      According to your comment “With so many correlations one might expect at least one positive correlation.” and your previous #2 comment, the sentence “On the other hand, regarding the observed inverse association between maternal cow’s milk intake during pregnancy and childhood emotional problems, p value for adjusted odds ratio between extreme quartiles was 0.0015 and p for trend was 0.003; α error is very small and thus this finding is not likely to be chance phenomenon.” was added to the Discussion section (Pages 7-8 Lines 258-262 in the revised manuscript).

      Regarding your comment “Secondly, there is very limited physiological support for the finding.”, we think that a description of physiological support for physiological support for the finding is provided in the Discussion (“Milk fat globule membrane (MFGM) and milk proteins might be particularly important. Postnatal supplementation of ganglioside- and phospholipid-enriched complex-milk-lipids beta serum concentrate improved learning and memory in rats [29]. Formula with bovine MFGM promoted reflex development and changed brain phospholipid and metabolite composition in rats [30]. A randomized controlled trial demonstrated that infants fed an MFGM-supplemented experimental formula between 2 and 6 months of age performed better on cognitive testing at 12 months compared with infants fed standard formula and at a level not significantly different from the breastfed reference group [31]. When the diet of postnatal piglets was supplemented with lactoferrin, an iron-binding milk glycoprotein, from days 3 to 38, lactoferrin supplementation promoted early neurodevelopment and cognition by upregulating the brain-derived neurotrophic factor signaling pathway and polysialylation [32]. A recent study in chronic stress model mice has revealed the potential of milk casein to prevent stress-induced brain dysfunction and anxiety-like behavior [33]. Given the protective effects of MFGM or any milk proteins on neurodevelopment in early childhood, higher maternal intake of cow’s milk during pregnancy might provide persistent beneficial effects against the development of emotional problems in children.”; Page 7 Lines 240-256 in the revised manuscript). We did not add any text concerning this issue.

Reviewer 5 Report

Well-done on addressing the comments. Please edit lines 86 - 88 as follows:

"Using the contact information in this form, research technicians gave each pregnant woman a detailed explanation of the KOMCHS by telephone and sent her a self-administered questionnaire after obtaining her agreement verbal consent.

Author Response

Reviewer 5 Comments for the Author...

Well-done on addressing the comments. Please edit lines 86 - 88 as follows:

"Using the contact information in this form, research technicians gave each pregnant woman a detailed explanation of the KOMCHS by telephone and sent her a self-administered questionnaire after obtaining her agreement verbal consent.

Response:

Thank you for your editorial comments. We have revised the manuscript in accordance with your remarks as follow: “Using the contact information in this form, research technicians gave each pregnant woman a detailed explanation of the KOMCHS by telephone and sent a self-administered questionnaire after obtaining verbal consent” (Page 2 Lines 88-90 in the revised manuscript).